# Process Parameter Optimisation for Endohedral Metallofullerene Synthesis via the Arc-Discharge Method

**Sapna Sinha [1],\*, Karifa Sanfo [2], Panagiotis Dallas [3], Sujay Kumar [4] and Kyriakos Porfyrakis [2],\***

1   McGovern Institute and K. Lisa Yang Center for Bionics, Massachusetts Institute for of Technology, Cambridge, MA 02139, USA
2   Faculty of Engineering and Science, University of Greenwich, Central Avenue, Chatham Maritime, Kent ME4 4TB, UK
3   Institute of Nanoscience & Nanotechnology, NCSR "Demokritos", Patr. Gregoriou E' & 27 Neapoleos Str., Agia Paraskevi, 15341 Athens, Greece; p.dallas@inn.demokritos.gr
4   Department of Physics, Babasaheb Bhimrao Ambedkar Bihar University, Muzaffarpur 842001, Bihar, India
*   Correspondence: ssapna@mit.edu (S.S.); k.porfyrakis@greenwich.ac.uk (K.P.)

**Abstract:** Fullerenes have a unique structure, capable of both encapsulating other molecules and reacting with those on the exterior surface. Fullerene derivatives have also been found to have enormous potential to address the challenges of the renewable energy sector and current environmental issues, such as in the production of n-type materials in bulk heterojunction solar cells, as antimicrobial agents, in photocatalytic water treatment processes, and in sensor technologies. Endohedral metallofullerenes, in particular, can possess unpaired electron spins, driven by the enclosed metal atom or cluster, which yield valuable magnetic properties. These properties have significant potential for applications in molecular magnets, spin probes, quantum computing, and devices such as quantum information processing,, atomic clocks, and molecular magnets. However, the intrinsically low yield of endohedral fullerenes remains a huge obstacle, impeding not only their industrial utilization but also the synthesis and characterization essential for exploring novel applications. The low yield and difficulty in separation of different types of endohedral fullerenes results in the usage of a large amount of solvents and energy, which is detrimental to the environment. In this paper, we analyse the methodologies proposed by various researchers and identify the critical synthesis parameters that play a role in increasing the yields of fullerenes.

**Keywords:** endohedral metallofullerenes; EMFs; arc discharge; bottom-up synthesis; in situ doping

## 1. Introduction

Carbon, the building block of life and the fourth most abundant element in the universe, is one of the most indispensable materials known to humankind. In addition to the two naturally occurring allotropes, graphite and diamond, carbon is also capable of taking several remarkable physical forms on the nanoscale. One such distinct allotrope in the family of carbon nanomaterials is the fullerene. Its discovery [1] sparked a great deal of curiosity and interest amongst the scientific community due to its unique closed, cage-like structure, composed solely from $sp^2$ hybridized carbon [2,3]. Within the first few years after the discovery of buckyball $C_{60}$, the term 'fullerene' achieved a uniform acceptance as the general name for these carbon cages. Fullerenes can have different numbers of carbon atoms, shapes, and symmetry; hence, the largely accepted notation system [4] is to subscript the number of carbon atoms to state the class of the fullerene. The structure of $C_{60}$, the most abundant buckyball, has been verified to consist of 12 five-membered rings and 20 six-membered rings [2]. Most of the fullerenes obey the isolated pentagon rule (IPR), where pentagons are exclusively surrounded by hexagons, which increases their stability. However, several fullerenes, especially a number of endohedral fullerenes (see next section),

do not follow the IPR rule [5]. Instead they follow the maximum pentagon separation rule, where they exhibit maximum pentagon separation for maximum stability [6].

Fullerenes are known to have different isomers, with $C_{60}$ having 1812 theoretically different isomers [7,8]. These isomers can have different symmetry, form, and energetic stability. An example of the different structures of the six most stable isomers of $C_{80}$ is shown in Figure 1, with $D_{5d}$ and $D_2$ being the most stable and abundantly found isomers [9]. Fullerene isomers are also known to have different oxidation potential and chemical reactivity. For instance, the $D_3$ isomer of $C_{78}$ is easier to oxidize than the $C_{2v}$ isomer of the same fullerene by about 0.25 V [10]. Another very intriguing feature of the fullerenes is that they have a hollow space, which is large enough to encapsulate other atoms, ions, molecules, and various kinds of clusters, giving rise to a new class of hybrid materials called endohedral fullerenes.

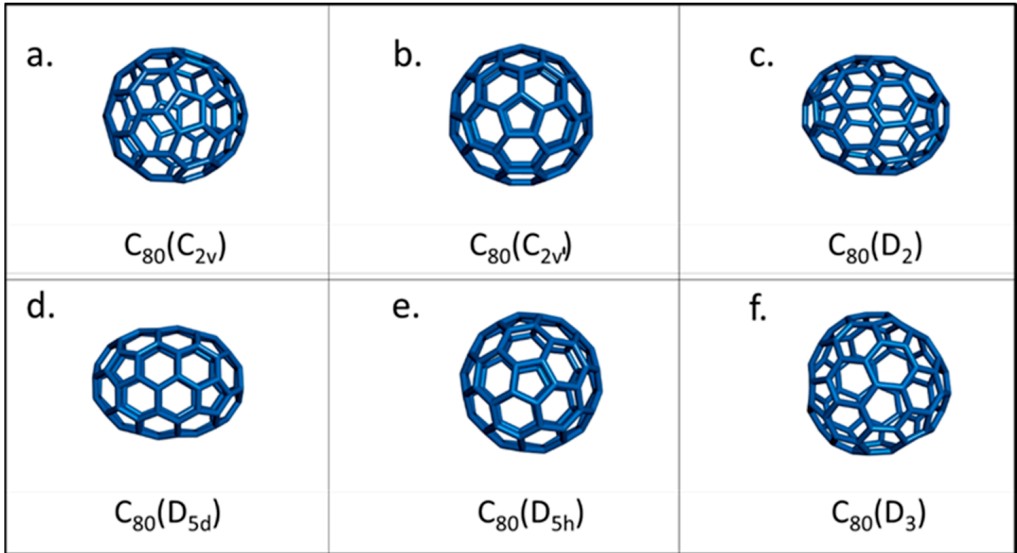

**Figure 1.** (**a**–**f**) Schematic representation of the different isomers of the $C_{80}$ cage.

## 2. Structure of Endohedral Fullerenes

The ability of different sizes of fullerene cages to incorporate different types of metals and clusters can lead to the formation of materials with novel solid-state properties. The robust nature of the carbon cage plays a very important role in isolating the atoms from the exterior environment, while the electronic properties of the cage can be tuned by the selection of the atoms inside. La@$C_{82}$ was among the first discovered endohedral fullerene, where the symbol @ denotes the inclusion of a La atom inside a $C_{82}$ cage. Since then, fullerenes encapsulating metallic [11], carbide [12,13], nitride [14], and many methano- [15] and oxide- [16] clusters have been synthesized and isolated over the years. In fact, fullerenes containing single atoms of elements that otherwise are not stable, such as noble gases [17–19] and non-metals [20] such as nitrogen [21], have also been synthesized. Transition metals and actinides were amongst the challenging elements to encapsulate within fullerenes, primarily due to their reactivity and the difficulties associated with their incorporation into the graphite rod for arc-discharge synthesis. However, over the past decade, significant progress has been made in also encapsulating various transition metals within fullerenes. [22] In these cases, while describing the diversity of the fullerene cages, prefixes and cage symmetries are also used. For instance, $Sc_4(\mu_3\text{-}O)_3@I_h(7)\text{-}C_{80}$ represents the endohedral cluster group, cage structure, and point-group symmetry, following the nomenclature of Fowler and Manolopoulos [4].

In 1991, researchers succeeded in encapsulating metal atoms, which opened a complete new set of possibilities in the field of material development, generally referred to as endohedral metallofullerenes [23]. Among the endohedral fullerenes, the trimetallic nitride-containing cages of the type $M_3N@C_{2n}$ (where M = Sc, Gd; 2n = 68–96), shown in Figure 2,

have been one of the most studied materials since their discovery in 1999 [14] due to their high yields during synthesis and their appropriate HOMO-LUMO levels, desired for applications in solar cells. However, despite the comparable high yields, empty fullerenes still constituted as major product during the synthesis process. The yields of $M_3N@C_{2n}$ were further improved by Dunsch and Yang [24], who introduced ammonia during fullerene synthesis and produced an endohedral fullerene, $Sc_3N@I_h$-$C_{80}$, as the majority product for the first time. This family of fullerenes has an especially high yield because of the charge transfer from the encaged cluster, which results in a stable ion pair and stabilizes the higher fullerene cages of various isomeric structures. In addition, the large HOMO-LUMO gaps give further kinetic stability to the hybrid molecule.

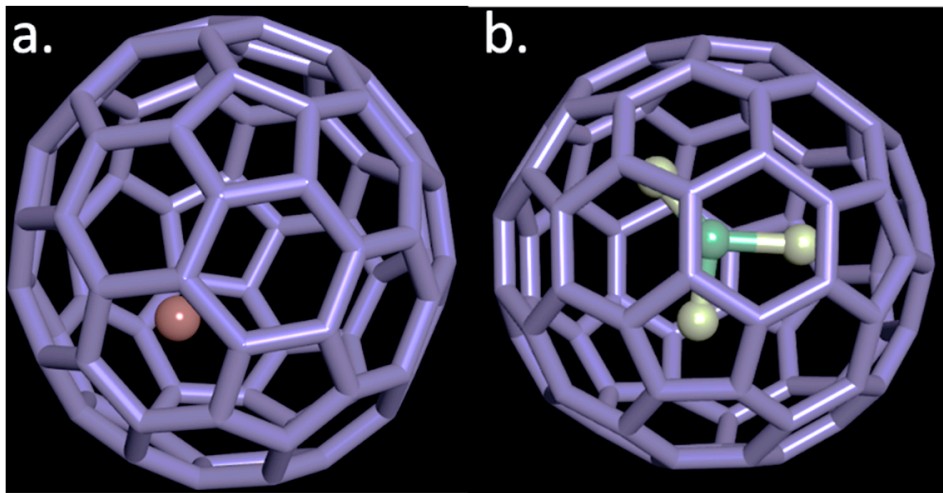

**Figure 2.** A representative figure of two endohedral metallofullerenes, (**a**) monometallic in a $C_{82}$ cage and (**b**) trimetallic nitride encapsulated in a $C_{80}$ fullerene cage. The red and the yellow balls correspond to the metal ion encapsulated in the carbon cage and the green to nitrogen.

### 2.1. Endohedral Metallofullerenes: EMFs

In the years following the discovery of $C_{60}$, the study of empty cage fullerenes heavily dominated the field of nanomaterials. The focus on empty cage buckyballs often resulted in the blind assumption that endohedral metallofullerenes (denoted as EMFs) possessed the same structural or chemical properties as the corresponding empty cages. However, over the years, after more studies were conducted on individual EMFs, it became established that they exhibit novel structural, chemical, and electronic properties, which were completely different to those of their parent empty cages. For instance, carbon cages of EMFs have different isomeric structures as compared to their parent fullerene cage, and they also show different solubility in individual solvents [25]. As for their electronic and chemical behaviour, their chemical reactions with radicals [26] and many other cyclo-addition [27] reactions show significantly different patterns and redox potentials [28]. For instance, the reduction potentials of $La@C_{82}$ and $La_2@C_{80}$ are 0.70 V and 0.81 V lower than that of $C_{60}$ [29]. The reason for such a huge difference is largely attributed to the covalent interaction between the enclosed species and the carbon cage as well as the charge transfer phenomenon from the metallic species to the fullerene cage.

The most studied metallofullerenes involve rare earth metals, such as La, Sc, Dy, and Tm [23,30–35], and main group metals such as Li and Ca [36]. There is an extensive array of EMFs that have one, two, three, or even four metal atoms trapped inside them [37]. Alternatively, they can also contain $M_3N$ [14] or $M_2C_2$ units [12], where M refers to metals. The key to understanding the electronic properties of these hybrid materials is through studying the degree of charge transfer from the metal species to the exterior cage, which alters its electronic state. While the number of electrons transferred from a majority of metal atoms to the cage is known and verified using X-ray diffraction, absorption, and

other spectroscopic methods [31–35], we observed that contradictory results still exist in the literature regarding the effect of the various parameters on the arc synthesis and the formation mechanism of these fullerenes. For instance, Dunk et al. claimed that the formation mechanism is governed by a charge transfer between the metal and the cage [38]. Through detailed mechanistic studies, they revealed that the $C_2$ insertion during the bottom-up synthesis becomes more difficult as the oxidation state of the metal increases. This charge transfer and the nature of the cluster inside the cage also dictates a series of properties, such as electrochemical behaviour and light emission. In fact, Ito et al. studied $Er_2@C_{82}$ and $Er_2C_2@C_{82}$, which showed nearly identical cage properties with a 6-negatively charged $C_{82}^{6-}$, using fluorescence spectroscopy [39]. Most notably, this cage charge contradicted an earlier publication by Wang et al. claiming that the $Er_2@C_{82}$ and $Er_2C_2@C_{82}$ compounds have a 4- negatively charged cage, i.e., $C_{82}^{4-}$ [40]. Furthermore, Ito et al. showed that the characteristic emission of the erbium ions at 1520 nm was influenced by the symmetry of the cage, an observation later echoed by Plant et al. [41] Until as recently as 2020, previously unknown factors, such as enantiomers and enthalpy, were discovered to play vital and evolving roles in the synthesis mechanism for fullerenes $C_{2n}$ ($50 \leq 2n \leq 70$) [42]. After a critical analysis of research articles [43,44], it can be concluded that the complete mechanism of their formation, either through a bottom-up or top-down approach, and the charge transfer itself remains elusive and highly controversial [45]. In fact, the insights derived from experimental results have also been contradictory to theoretical results in the field. For instance, it has been suggested that factors such as the $C_2$ concentration and temperature may dictate the more favourable direction of the reversible $C_2$ injection/ejection reaction during the synthetic process [46]. However, theoretical studies have highlighted that the $C_2$ injection is more energetically favourable than its $C_2$ ejection counterpart, contradicting the experimental data on the importance of temperature and concentration [47,48]. The missing key links and low synthesis yield of fullerenes make it difficult to investigate the fullerene formation mechanism. Utilizing both experimental and theoretical frameworks is therefore necessary to answer some of the fundamental questions of this field.

As briefly mentioned in the previous section, the isomeric structures of the carbon cages encapsulating metals differ from their empty counterparts [34]. Further study of the positions and movements of the encapsulated atoms is also important in determining the chemical and physical properties of EMFs, including their magnetic behaviour, electron paramagnetic resonance (EPR), and electrochemical properties [49]. The structure of the EMFs themselves had long been speculated, but the first crucial evidence came through in 1995 when Shinohara et al. experimentally confirmed that the metal atom is indeed encapsulated within the fullerene cage [31]. The synchrotron X-ray powder diffraction patterns of $Y@C_{82}$ and the maximum entropy method (MEM) revealed that the yttrium atom was displaced from the centre and instead tightly bound to the cage [50]. The study also confirmed strong dipole–dipole and charge transfer interaction between the different molecules [49]. MEM/Rietveld analysis has long been used to determine the structures of metallofullerenes [31,50]. However, the results (structure and position of metals inside the carbon cage) obtained by this method have been in question in recent years. This started when Wang et al. [12] discovered that the scandium carbide endohedral fullerene, $Sc_3C_2@C_{80}$, was, in fact, what had long been considered to be a trimetallofullerene $Sc_3@C_{82}$ [51]. These incoherences emphasize the importance of $^{13}C$ NMR and synchrotron X-ray structural analysis in determining the structures and encapsulated cluster inside the fullerene cage and suggest the unreliability of the decades of results obtained exclusively using MEM/Rietveld analysis when determining the structures of metallofullerenes [52–54]. Consequently, the structures and positions of metals inside a fullerene have been re-examined in the past few years [55–58]. Furthermore, it was also observed that these metal clusters had motions and trajectories inside the cage [59], which also changed positions depending on the temperature [60,61].

## 2.2. Environmental Application

A massive number of publications in fullerene-related research show the possibility of using fullerenes for a variety of environmental and energy applications. The characteristics of fullerenes that give them an advantage over other materials include their robust carbon structure, absorption of light in the visible range, electron-accepting capabilities, and more recently, superconductivity [62]. All these properties make fullerenes especially sought-after materials for energy applications, especially organic solar cells [63]. They have become, and remain, an important part of high-performance organic solar cells.

## 2.3. Synthesis

Over the past few decades, EMFs have been proposed for a number of applications, including energy harvesting in organic solar cells [64] and MRI contrast agents [65–67]. EMFs have been synthesized through a range of different methodologies, such as laser ablation of graphite [23,68], arc discharge [30,33], ion bombardment [36,69], resistive heating [70], and electron initiation inside carbon nanotubes (CNTs) within aberration-corrected high-resolution transmission electron spectroscopy (AC-HRTEM) (CNT acts as a nanoreactor container to facilitate the formation of endohedral fullerenes from metals and the amorphous carbon atoms trapped inside. The process is energetically enabled under the presence of an electron beam inside an AC-HRTEM.) [71]. Some of these popular methods are summarized in Table 1 below.

**Table 1.** Advantages and disadvantages of different synthetic approaches.

| EMF Synthesis Method | Advantages | Disadvantages |
|---|---|---|
| Arc discharge [72] | • Low setup and maintenance costs <br> • Fewer structural defects <br> • Access to the broadest catalogue of EMF | • Difficult to scale-up <br> • Requires extensive purification <br> • Low yield (1–30%) |
| Laser ablation [72] | • Higher purity of products <br> • Useful for mechanistic studies | • High cost for reactor building and maintenance |
| Molecular surgery [72,73] | • Low energy impact <br> • Simple HPLC purification | • Several synthetic steps required to open/close the fullerenes cage <br> • Harmful synthetic solvents <br> • Limited to non-metal and alkali-metal EMFs <br> • Low yield (~0.1 to 1%) |
| Ion implantation [72] | • Lower pressure required (5–10 mbar) <br> • Low-temperature process, where ions are accelerated towards the carbon cages <br> • Simple HPLC purification possible | • Limited to non-metal and alkali-metal EMFs <br> • Very low yield ($10^{-5}$ to $10^{-4}$%) |

The Kratschmer–Huffman arc method has been widely used and accepted as the most efficient and cost-effective technique to produce EMFs and, hence, is the focus of this review. Figure 3a shows a schematic diagram of a typical standard direct current (D.C.) arc-discharge apparatus, and Figure 3b shows the actual image of one in the Porfyrakis laboratory in the Department of Materials at the University of Oxford.

While the Kratschmer–Huffman method immediately became the most popular choice of EMF synthesis, it was not exempt from certain limitations. Most notably, only 60–70% of the graphite electrode was transferred to the carbon condensate (CC) containing the fullerenes and EMF, while the remaining 30–40% was lost as graphite build-up on the cathode. This disadvantage has now been eliminated by utilizing a high frequency arc discharge to supply the arc. The symmetrical setup of the electrodes allowed reaching up to a 100% conversion of the electrode material into the fullerenes and EMF containing CC [74]. Additionally, high-frequency arc discharge can also be used for further mechanistic investigation of fullerene synthesis.

A metal oxide/graphite composite is used as the anode, whereas graphite is used as the cathode, and both of them are arced in the D.C. mode in the presence of an inert gas (usually helium). Despite three decades of extensive research and various proposed pathways, a well-established formation mechanism for the synthesis of fullerenes and EMFs is still missing. Their ambiguous reaction pathways, alongside the violent conditions involving the plasma formation, arc, and chemical ionization that are required for the synthesis, are a major drawback in studying the synthesis mechanisms of these unique carbon nanostructures. Furthermore, a few other drawbacks hinder the scaling-up of their synthesis and consequently the applications of endohedral metallofullerenes. The overall yield of the arc-discharge synthesis of EMFs is low, with the ratio of EMF-to-empty-cage typically at around 1% [49]. As a result, a mixture of empty cage and endohedral fullerenes is typically produced, thus limiting the applications of the existing chromatographic methodologies for their fast and efficient separation. This results in the use of excessive amounts of solvents to efficiently separate them, which is neither ecological nor environmentally friendly. Therefore, it becomes important to review the synthesis of fullerenes. Shinohara published the first detailed scientific review of EMF research in 2000 [75], followed by an extensive update on endohedral fullerenes by Popov et al. in 2012 [76]. Many other groups have addressed the synthesis and properties of endohedral fullerenes [76–79], including nitride fullerenes [24], carbide cluster fullerenes [80], and oxide cluster fullerenes. Herein, we focus primarily on critically discussing the key findings in the field for optimizing of synthesis parameters of endohedral metallofullerenes and the new chemically activated method, with a particular emphasis on the post-2000 science along with comparisons to the early methodologies. The yield of the EMF has been observed to be highly dependent on the pressure of He gas, the arc current, the arc gap, and the composition of anodes, among other factors [51,81]. Liu [82] and Bandow [83] have argued that other parameters, such as the presence of catalysts (Cu, Fe, etc.), the back-burning of anodes, and in situ activation during arc discharge, also have an effect on the EMF yield. However, the latter two parameters would not increase the yield by a huge fraction because they do not modify the arc conditions drastically or change/influence the synthesis chemistry, e.g., lower the activation energy of the formation of the endohedral metallofullerenes. However, in order to maximize the efficiency of the method, a balance between the time spent during the synthesis and increase in yield needs to be established. Predominant aspects of overcoming the above challenge are anode composition, ionization potential, catalysts, and helium pressure in synthesis [74].

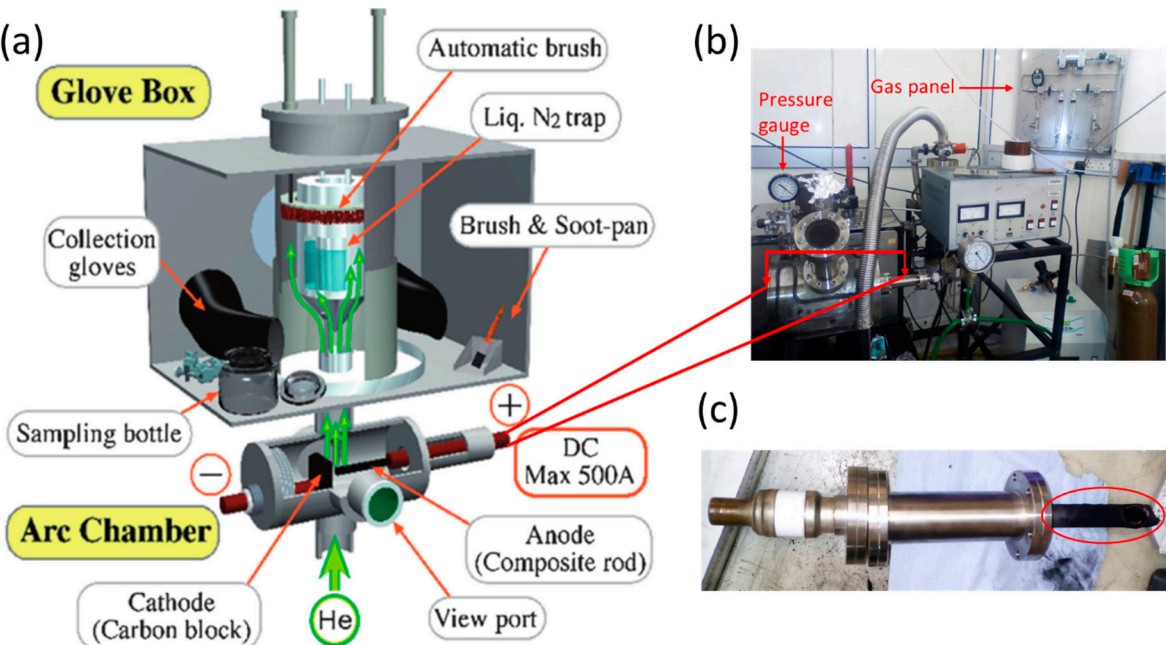

**Figure 3.** (**a**) A schematic diagram of a typical arc-discharge apparatus used for synthesizing fullerenes (Reprinted/adapted with permission from Ref. [84]. Copyright (**c**) 2007 The Japan Society of Applied Physics). (**b**) An image of an actual arc-discharge apparatus from the Porfyrakis laboratory at the University of Oxford. (**c**) Image of a metal oxide/graphite composite rod used during an arc-discharge process.

In recent years, the Kratschmer–Huffman arc design has also been modified mechanically in several ways to make it more environmentally friendly. For instance, the use of demineralized coal electrodes, minimizing the use of current and using direct current instead of high alternating current are a few ways in which the process has been made more efficient and less energy intensive [85]. Recently, Kyesmen et al. showed scaled-up production yield of fullerenes by employing such methods and including efficient energy usage by using resistive heating of one of the electrodes along its length [86]. Such green methodologies should be further explored for future research purposes and industrial-scale production of fullerenes to ensure an environment friendly approach towards using nanomaterials.

## 3. Reaction Parameters and Pathways

In this section, we will delve into the critical parameters that are essential for achieving the highest yield of EMFs, with a special focus on encapsulation in the $C_{82}$ cage.

### *3.1. Anode Composition*

Metal oxide (e.g., $La_2O_3$, $Y_2O_3$, etc.) mixed with graphite has generally been used as the anode material during the arc-discharge process. However, there has been no general consensus on the ratio of metal/carbon used in the metal. Inakuma et al. also showed that the abundance of different types of EMF present in the soot depends heavily on the metal/carbon mixing ratio in the composite rods [51,81]. The M/C ratio has varied widely over the years, also depending on the metals that are incorporated inside the graphite, as broadly summarized in Table 2. Furthermore, it has also been found that anodes made out of metal-carbide rods actually produce EMFs in a bigger yield than metal-oxide rods, which has been experimentally proved by several authors [82,83]. Using La as the example metal, the authors showed that the presence of $LaC_2$ in supersaturated carbon vapor plays an important role in generating the metallofullerenes. However, this conclusion is in contrast with the work published later by Lian et al. [87], who studied Ni-La alloy anode rods for the preparation of lanthanum-containing metallofullerenes (La@$C_{82}$ (I), La@$C_{82}$ (II), and $La_2$@$C_{80}$). It was found that $LaNi_2$ is a much better dopant than conventional $La_2O_3$ in the

anode rod, and the ratio of La:C is very important for increasing the yield. The absence of oxygen in the LaNi$_2$ dopant can partly be the reason for such observations as well as the possible catalytic effect of Nickel in the formation of fullerenes. The authors also found that the use of LaNi$_2$ instead of La$_2$O$_3$ enhanced the amount of raw soot produced from arc discharge. The authors attributed the results to reduction in the amount of formation of cathode deposits and debris. The optimum conditions increased the yield of La-EMF by almost 30% compared to conventional substrates. Comparing the existing literature in this field, as summarized in Table 2, it seems rather hard to deduce which single metal/carbon ratio in the anode would produce the best yields for different types of metallofullerenes. It surely is a huge determinant of the yield of a particular type of EMF, but more research, both experimental and theoretical, needs to be undertaken in this direction to test different combinations of metals for anodes and also to explore the chemistry involved.

### 3.2. Ionization Potential

The ionization potential (I.P.) of the EMFs has been known to play a very important role in EMF extraction and chemical reactivity, but it only recently came to light that the first ionization potential of the metal itself plays an important role in EMF synthesis. Ross et al. deduced that the I.P. of the metal atom not only influences the efficiency of the synthesis but may partly determine the metallofullerene formation [88]. Their group noted that when graphite anode rods were impregnated with Y/La atoms and with Sc/Y atomic composition in equal ratio, there was a greater abundance of La- and Y-containing metallofullerenes than Y or Sc, respectively. The results suggested that metals with lower I.P. (La < Y < Sc) tend to react more efficiently, i.e., the growth reaction of metal ions and fullerene precursor species plays an important role in the arc-discharge process. This is consistent with other experimental studies that show that lanthanides are easier to encapsulate inside a fullerene cage compared to other alkali earth metals or transition metals [89]. Comparing the yields of the synthesis of Pt-EMF [90,91], Fe-EMFs [92], and Sc-EMFs (Pt $_{I.P.}$ = 8.96 > Fe $_{I.P.}$ = 7.90 > Sc $_{I.P.}$ = 6.56) [93], Pt-EMFs have the lowest yield, whereas the Sc-EMFs have the highest yield, even though the covalent radii of the metals are not very different (Pt = 1.17 A; Fe = 1.3 A; Sc = 1.44 A). Since the overall yield of EMFs is generally low, and their synthesis is a very time-consuming process, it is important that the choice of the metal atom, which can potentially form EMFs, is based on their ionization potential and chemical nature of bonding with the carbon clusters.

### 3.3. Effect of Catalyst

The catalytic growth of carbon nanomaterials has long been known from the synthesis of carbon nanotubes [94], and more recently, graphene [95]. However, the use of catalysts for the synthesis of fullerenes was not developed until 2008, when Stevenson et al. [16] doped the graphite rods with copper nitrate in a modified arc chamber. They observed that the presence of metallic copper increased the overall yield of endohedral fullerenes. It still remains unclear if it also facilitates the formation of endohedral mono-metallofullerenes and if the ratio of metallofullerenes to other empty cages in the carbon soot is, in fact, higher than previously reported. Other groups have noted that using transition metal alloys (e.g., Ni) in the anode composition [87] increases the yield of the EMFs. But again, results published in the literature make it difficult to judge whether the transition metal plays an active role as a catalyst or if it just reduces the detrimental effects that oxygen has, whist using metal oxide composite rods during the arc discharge. We deduce that catalysts have not been studied in depth largely because of the difficulty in producing homogeneous composite anode rods containing different catalysts such that they can be actually used systematically during the arc operation. The development of new methods for efficient dispersion of catalytic particles inside the electrodes during their fabrication could help solve these problems. Studying the effect of metal clusters or metal alloys as additives in the arc discharge, could be worthwhile not only for increasing the yield of EMFs but

also for elucidating the formation pathways for fullerenes and carbon nanotubes in the arc reactor chamber.

### 3.4. Effect of Helium Pressure

The presence of helium in the arc-discharge chamber is well known to be beneficial for the formation of endohedral fullerenes [96]. It is still not clear what the exact mechanism behind it is, but it is believed that He, being a small and inert species, facilitates collisions between carbon atoms without taking part in reactions. Therefore, it enhances the possibility of carbon–carbon bond formation. There is also a general consensus that varying the pressure of helium leads to different yields of the EMFs. A literature survey (Table 2) in the field shows that using lower pressures of helium (50–100 mBar) has been the standard arc-discharge parameter until recently, when researchers started using high He pressures (up to 500–700 mBar) for EMF synthesis. In 2017, Qian et al. [97] published the first theoretical study on how the self-assembly of fullerene during the arc-discharge process is influenced by the presence of helium. They also proposed how the concentration of helium can change the distribution of different fullerene formation in the arc. Similar results were shown for synthesizing fullerenes in a high-frequency arc-discharge system, as reported by Churilov et al. [74]. The authors experimented with a range of helium pressures (30–150 kPa) and their effect on the formation of different types of Y-EMFs, and concluded that 120 kPa results in the maximum yield for the EMF $Y@C_{82}$. Moreover, empty cages were found to increase with the helium pressure but not the EMFs. The effect of helium pressure in both the traditional arc discharge as well as high-frequency arc discharge suggests a difference in the mechanism of formation of the empty cages and the EMFs. Further studies in this area can help optimize the exact synthetic parameters for the desired type of fullerene. Another helium pressure-dependent study was reported by Cherviakova et al. [98], where they carried out a series of experiments by keeping all the variables constant and only changing the He pressure in the chamber from 30 to 360 mBar. The experiment was carried out for various EMFs, and they found that although 120 mBar helium pressure yields the optimal condition for synthesis of different $M@C_{82}$, there is a huge variation in the obtained optimal yield. For instance, although both $Sc@C_{82}$ and $Gd@C_{82}$ showed the best synthetic yields at 120 kPa He pressure, the yields were very different, with $Sc@C_{82}$ having 9.6 wt% ratio of all fullerenes, as compared to $Gd@C_{82}$, which had only 5.4 wt%. However, there is a massive variation in homemade arc-discharge apparatus as well as differences in the exact operational parameters used for experiments carried out at different places. Therefore, there is a need for a systematic study on a single apparatus, for synthesizing the range of EMFs and determining if the pressure conditions vary depending on the entrapped metals. Moreover, a wider pressure range should also be tested so that the exact reaction pathway behind the formation of different types of EMFs can be properly investigated.

**Table 2.** Summary of the parameters used over the years to synthesize different types of endohedral metallofullerenes.

| Year | Anode | Fullerene | He Pressure (mBar) | Current | Extraction Solvent | References |
|------|-------|-----------|--------------------|---------|--------------------|-----------| 
| 1992 | $Sc_2O_3$ | Sc-EMF | 66.6 | - | Toluene and Pyridine and $CS_2$ [1] | [35] |
| 1992 | $Sc_2O_3$ | Sc-EMF | 266.6 | - | Toluene | [32] |
| 1992 | $La_2O_3$ | La-EMF | 266.6 | - | Toluene | [33] |
| 1993 | $Sc_2O_3$ M/C = 2.3/100 atomic ratio | Sc-EMF | 66 | 220 A | $CS_2$ | [81] |
| 1993 | 8 wt% $La_2O_3$ | $La@C_{82}$ | 133.3 | 2.3 A/mm$^2$ | Toluene | [83] |
| 1994 | $Sc_2O_3$ M/C = 1:10 | | 66–133 | 500 A | $CS_2$ | [51] |

**Table 2.** *Cont.*

| Year | Anode | Fullerene | He Pressure (mBar) | Current | Extraction Solvent | References |
|------|-------|-----------|--------------------|---------|--------------------|-----------|
| 1995 | $La_2O_3$ (La/C = 2:100) | $La@C_{2n}$ | 133 | 200 A | Toluene | [82] |
| 1995 | $Gd_2O_3$ (1–2% atomic%) | Gd-EMF | 60 | 250 A | TCB [2] | [99] |
| 1995 | - | $Y@C_{82}$ | 66 | 500 A | Toluene | [31] |
| 1996 | $CeO_2$ | $Ce@C_{82}$ | 66 | | DMF | [100] |
| 1996 | Tm/C = 1:50 | $Tm@C_{82}$ | 200 | 175 A | Toluene | [34] |
| 1996 | $Pr_6O_{11}$ (Pr/C = 1:100) | $Pr@C_{82}$, $Pr_2@C_{80}$ | 66 | | DMF [3] | [101] |
| 1998 | - | $Sc@C_{82}$ | 66 | 300–400 A | Toluene | [93] |
| 1999 | $Sc_2O_3$ Sc/C = 2.8/100 | $Sc_3@C_{82}$ | 66 | 500 A | $CS_2$ | [11] |
| 2002 | M/C = 1/100 (M = La,Y) | $La@C_{2n}$ [4], $Y@C_{2n}$ | 160 | 90 A | o-xylene+ DMF/DMA/DMSO | [102] |
| 2002 | Tm/C composite rod | $Tm@C_{82}$ | 240 | 600 A | TCB | [103] |
| 2002 | | $Gd@C_{82}$ | 960 | 50 A | DMF | [104] |
| 2003 | $Gd_2O_3$ | $Gd@C_{60}$ | 133 | - | Dichlorobenzene | [105] |
| 2004 | $MNi_2$ | All EMF | 960 | 50 A | DMF (~6–7%) $CS_2$ (~1.5%) Pyridine (~1.5%) Aniline (~1%) [5] | [106] |
| 2005 | $CeO2$ (Ce/C = 2/100) | $Ce_2@C_{80}$ | 66.6 | 70 A | TEA/acetone [6] | [107] |
| 2006 | $Sc_2O_3$ Sc/C = 2/100 | Sc- EMF | 66 | 150 A, 40 V | TCB | [108] |
| 2010 | $Gd_2O_3$ | $Gd@C_{82}$ | 200 | 90 A | DMF liq./liq. extraction | [109] |
| 2014 | $Gd_2O_3$ | $Gd@C_{82}$, $Gd@C_{84}$ | 506 | 110–120 A | o-xylene and DMF | [110] |
| 2017 | $Gd_2O_3$ (M:C = 1:1) mass ratio | $Gd@C_{82}$ | 1200 | 50–400 A | $C_5H_5N$ | [98] |
| 2017 | $Sc_2O_3$ (M:C = 1:1) mass ratio | $Sc@C_{82}$ | 1200 | 50–400 A | $C_5H_5N$ | [98] |
| 2017 | $Er_2O_3$ (M:C = 1:1) mass ratio | $Er@C_{82}$ | 1200 | 50–400 A | $CS_2$ | [98] |
| 2017 | TiO (M:C = 1:1) mass ratio | $Ti@C_{82}$ | 600 | 50–400 A | $CS_2$ | [98] |
| 2017 | $Y_2O_3O$ (M:C = 1:1) mass ratio | $Y@C_{82}$, $Y_2@C_{82}$ | 600 | 50–400 A | $CS_2$ | [98] |
| 2017 | $Y_2O_3O$ (M:C = 1:1) mass ratio | $Y@C_{82}$ | 1200 | 50–400 A | $C_5H_5N$ | [98] |
| 2023 | $Tb_4O_7$ (M:C = 1:15) molar ratio | $Tb@C_{82}$ | 400 mbar | 100 A | of $CS_2$/toluene | [111] |

[1] Carbon Disulfide; [2] 1,2,4-Trichlorobenzene; [3] N,N-Dimethylformamide; [4] 68 ≥ 2n ≥ 96; [5] Extraction yield by weight percent of (Tb, Y, La, Gd)-EMF to soot; [6] Triethylamine.

*3.5. Chemically Activated Arc Discharge*

3.5.1. In Situ Doping

Another way to address the low yield and the unwanted mixture of various types of fullerenes is by changing the reaction conditions through in situ doping of the graphite/metal electrodes, which are to be vaporized during the arc discharge. These methodologies require the incorporation of reactive gases and/or solid compounds that alter the conditions and influence the reaction equilibrium towards the formation of higher cages and, ideally, EMFs. One of the most typical examples is the use of reactive gases, e.g., $NH_3$ [112], whose incorporation leads to the scaled-up synthesis of trimetallic nitride fullerenes, for instance, making the $Sc_3N@I_h$-$C_{80}$ the third most abundant fullerene, after $C_{60}$ and $C_{70}$. Another alternative is the doping of the graphite electrodes themselves with solid organic compounds. Core drilled graphite rods have been repeatedly doped with active inorganic or organic compounds. These compounds could be calcium cyanamide [113], urea and, most importantly, inorganic nitrate compounds. In that direction, Stevenson et al. pioneered the CAPTEAR method (the name stands for "Chemically Adjusting Plasma Temperature, En-

ergy and Reactivity") (Figure 4). They core-drilled and filled the graphite/$Sc_2O_3$ electrodes with copper nitrate powder and in a later study with metallic Cu, in order to study the reactive species generated during the synthesis procedure [114,115]. The decomposition of copper nitrate led to an exothermic reaction and the generation of reactive nitrogen gases, oxygen, and vaporized copper. A detailed analysis of the decomposition products can be seen in the references herein. These factors influenced the temperature, energy, conductivity of the arc, and reactivity of the plasma. Stevenson et al. observed that this particular arc discharge reaction leads to the exclusive and highly selective formation of TNT-type metallofullerenes ($Sc_3N@I_h$-$C_{80}$) and that the empty cage fullerenes were formed with a low yield. In their pioneering 2007 publication, they observed that this preferential formation of EMFs, based on HPLC calculated yield, was about 80% wt as compared to standard yield of 5–30 wt%. They observed that from the 13 mg of extracted fullerenes, 12 mg were identified as $Sc_3N@I_h$-$C_{80}$.

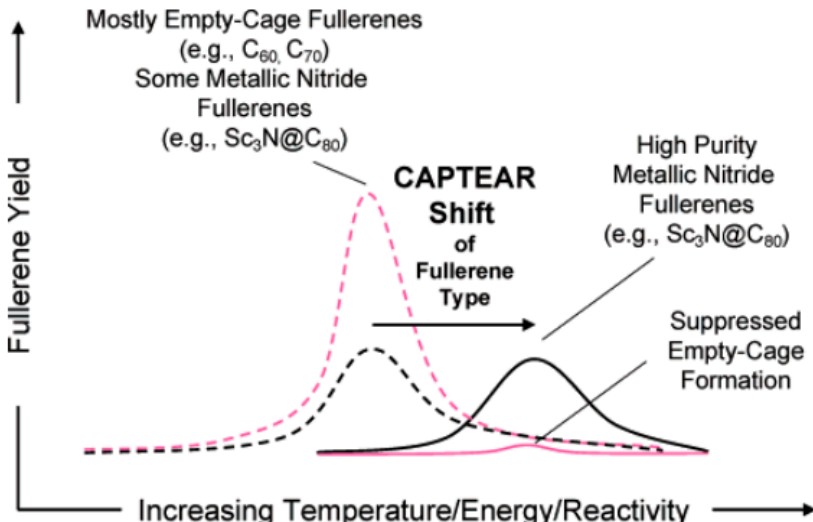

**Figure 4.** Parameter optimization of the CAPTEAR method to synthesize the desired endohedral fullerene. (Reprinted/adapted with permission from Ref. [114]. Copyright © 2007, American Chemical Society).

Expanding this method further, we recently developed a similar methodology for the study of the synthesis of gadolinium metallofullerenes and carbon nanotubes. We constructed phase diagrams regarding the effect of pressure and doping levels with copper nitrate on the overall yields. Figure 5 shows a phase diagram (reproduced from [116]) summarizing the ratio between the various fullerene families, depending on different reaction conditions. Another interesting aspect of this method is that when high doping with copper nitrate is applied, the yield of $C_{60}$ and $C_{70}$ appears to be inversed. During a conventional arc discharge reaction, the ratio of $C_{60}$ to $C_{70}$ is 3:1; however, under high levels of chemical doping, $C_{70}$ appears to have a higher yield compared to $C_{60}$. The results can be attributed to the difference in thermodynamic stability of these two fullerenes.

The thermodynamic parameters for the formation of $C_{60}$ were determined long after its initial discovery, mainly due to the early difficulties in producing sufficient amounts of the material and the low purity. Interestingly, despite the fact that the fullerene is a strained molecule and hence has much higher energy in free form as compared to diamond and graphite, it can still be formed in sufficient quantities. Higher cages than $C_{60}$ have a different structure and symmetry, with $C_{70}$, the second most abundant fullerene to exceed $C_{60}$, has a binding energy per carbon atom of more than 0.02 eV greater than $C_{60}$, whose binding energy is 7.40 eV/C [117]. Despite the thermodynamic favour of $C_{70}$, $C_{60}$ is kinetically favoured. According to previous reports, the heat of formation for $C_{70}$ is 2555 kJ/mol, and 2327 kJ/mol for $C_{60}$ [118]. The comparison between $C_{60}$ and $C_{70}$ is shown in Figure 6.

Similarly, fullerenes with more carbon atoms and EMFs themselves, which exhibit different isomers with different strains, will give a more complicated picture for determining the thermodynamic parameter of their synthesis [119].

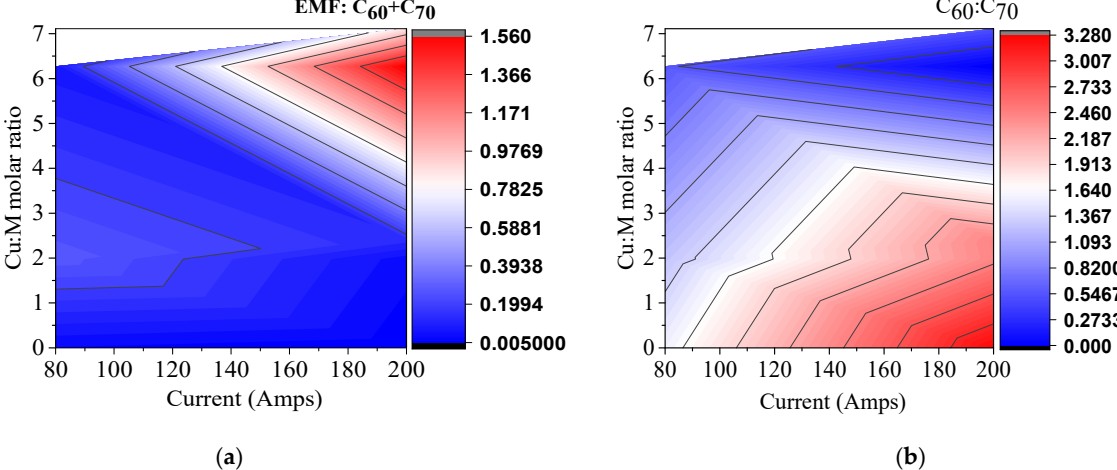

**Figure 5.** (**a**) The ratio of the higher fullerenes (>$C_{88}$) and EMF to the abundant $C_{60}/C_{70}$ vs. the current and Cu-to-rare-earth-metal molar ratio. (**b**) The ratio of $C_{60}$ to $C_{70}$ vs. the current and Cu-to-rare-earth-metal molar ratio (Reprinted/adapted with permission from [116]. Copyright 2006, Royal Society of Chemistry).

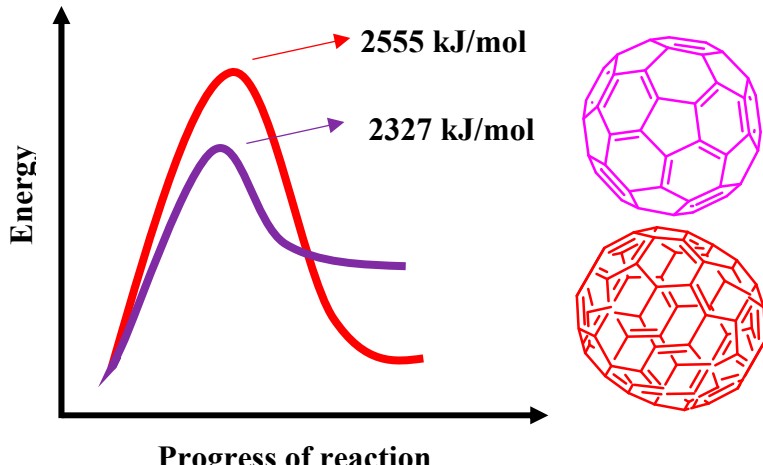

**Figure 6.** Kinetic vs. thermodynamic behaviour of the two most abundant fullerenes $C_{60}$ (in purple) and $C_{70}$ (in red). Reproduced from [118].

### 3.5.2. Effect of Reactive Gases: Methane and Ammonia

Chemically activated arc discharge presents a novel route not only for the control of the reaction pathway and hence the overall yield of the fullerenes, but also for the synthesis of novel EMFs with exciting properties, simply by carefully tailoring and studying the reaction conditions. Chen et al. [120] obtained different yields of dysprosium sulphite EMFs, a class of materials with interesting magnetic properties, by changing the levels of methane pressure used during synthesis. The dysprosium disulphide EMFs exhibit unique magnetic properties, with three energy barriers at 10.5, 48, and 1232 K. In general, a variety of magnetic behaviour can be observed in EMFs depending on the type of the incarcerated cluster. For example, the carbide $Er_2C_2@C_{82}$ behaves as a paramagnet, while the monometallic $Er@C_{82}$ shows an antiferromagnetic behaviour. An important dimetallic EMF, $ErSc_2N@C_{80}$, has even shown a long life of photoisomerization, which could be measured for 12 h at 20 K without any degradation. The synthesis of these EMFs is highly

important, as they hold promising applications as molecular memory elements [121] and materials for spintronics and telecommunications device applications. EMFs comprising erbium and scandium are of special interest, as erbium has important optical properties (1.52 μm emission at the telecommunications window), and scandium atoms have large nuclear spin ($I$ = 2/7), which is suitable for spin probe and atomic clock applications.

For the efficient synthesis of dysprosium sulphide EMFs, methane has been a valuable addition during the arc-discharge synthesis of metallofullerenes as a reactive gas, similar to the use of ammonia during the synthesis of fullerenes containing trimetallic nitride [122,123]. Popov demonstrated that the simple addition of 20 mbar $CH_4$ led to significant reduction in the formation of empty cage fullerenes during the synthesis of dysprosium sulphide fullerenes. The reaction mechanism in both methane- and ammonia-mediated synthesis is dominated by the hydrogen atom present in the gases. Hydrogen suppresses the growth of empty cage fullerenes and enables the more selective synthesis of mixed-metal and trimetallic nitride EMFs. On the other hand, other additives may have detrimental effects on the empty cage/EMF ratio. For example, during the graphite/metal oxide arc evaporation in the presence of $SO_2$, the empty fullerenes are the main products.

Furthermore, the nitrogen-containing atmosphere may lead not only to carbon cages but to a different class of fullerenes, the azafullerenes, where a nitrogen replaces a carbon atom and inherits other interesting properties. One of these azafullerenes, $Gd_2@C_{79}N$, exhibits a long spin relaxation time and a high magnetic spin state and is suitable for SQUID and pulsed EPR studies. In the case of dimetallic azafullerenes, gadolinium appears to have a higher yield compared to yttrium [124]. The gadolinium atoms couple in a ferromagnetic manner.

The formations of carbon nanotubes and metal carbides are two other reactions that take place during the arc vaporization and could be used as another observation to support either the bottom-up or the top-down mechanism for fullerene synthesis. Regarding the proposed mechanism of the charge transfer between the metal and the cage that stabilizes the EMF, see the work by Shinohara and Kroto [38].

In another example, Shinohara managed to isolate the missing low band gap endohedral metallofullerenes simply by adding Teflon tubes inside the arc reactor, in a short distance from the centre of the arc. During the arc-discharge process, the Teflon decomposed to $CH_2$ and $CF_3$ reactive groups that functionalize, in situ, the surface of the fullerenes. The resulting perfluorinated EMFs were more stable than their pristine counterparts, which are notoriously difficult to isolate with any common solvent [125]. Examples of the isolated fullerenes are $Y@C_{70}(CF_3)_3$, $Y@C_{72}(CF_3)$, $Y@C_{74}(CF_3)$ (I), $Y@C_{74}(CF_3)$ (II), and $Y@C_{74}(CF_3)_3$, as identified through MALDI-TOF analysis. The authors performed optical and DFT studies on these following analysis of other missing, small band gap EMFs. By further expanding the gallery of doping agents and mapping the temperature and pressure conditions, for example, by incorporating thermal and light sensors in the reactor and identifying the ideal molar ratio among the compounds, it is expected that the hindrances regarding the fullerene applications will be overcome. For example, in the case of nitrogen-based doping compounds, the reactivity in a CAPTEAR reaction appears to be based on the specific nitrogen anion, further demonstrating the complexity of the reaction [126].

## 4. Solvent Extraction

Solvent extraction is the next most important step in extraction of the EMFs from the carbon soot. Across much of the literature, the separation of fullerenes and/or EMFs from the soot is typically achieved by the Soxhlet extraction (SE) method [127]. Hence, the yield of extraction is dependent on the type of soot, the number of extraction cycles, and the solvents circulated through the sample. As discussed in the introduction, the metal inside the EMFs is not located in the centre of the carbon cage and is shifted towards one of the ends; therefore, this imparts a dipole moment to the EMF. Thus, the extraction of EMFs from the soot also requires a solvent of higher dipole moment, such as DMF, aniline, or DMSO, instead of toluene or other solvents of low dipole moments that have traditionally

been employed for extracting empty cages [110]. One of the common ways of extracting the maximum amount of fullerenes is by involving a two-step extraction method, where first a low-polarity solvent is used to extract the empty cages (>60%), followed by a high-polarity solvent for maximum extraction of the EMFs [110]. It is also important to note that different solvents work for different types of EMF. For instance, under constant parameters, Chervyakova et al. [98] showed that $CS_2$ could efficiently extract 11 wt% of $Y@C_{82}$ out of all fullerenes, compared to 5.0 wt % and 2.3 wt% yield of $Sc@C_{82}$ and $Gd@C_{82}$, respectively. Moreover, although $CS_2$ and $C_5H_5N$ are both considered a good solvent for extracting EMFs, it was observed that the overall yield of $M@C_{82}$ could vary widely between the types of solvents used under different synthetic parameters. For instance, at 360kPa He pressure, $CS_2$ extraction yielded about 12 times more $Sc@C_{82}$ than the use of $C_5H_5N$. Whereas at 120kPa, $CS_2$ extraction resulted in a threefold decrease in the $Sc@C_{82}$ yield as compared to using $C_5H_5N$. The authors have not commented on these observations, but perhaps a detailed study in this direction could help understand the relationship between synthesis processes and the resultant complex reactions of fullerenes as well as the corresponding methods to extract them efficiently using a particular solvent.

It has also been reported that porous glass, used for holding the soot and the solvent, increases the area of contact of the soot with the solvents and thus leads to a two- to threefold increase in the efficiency of the extraction process [110].

The latest methods and devices for the extraction of fullerenes have seen a shifted focus, aiming to address the limitations of the Soxhlet extraction method instead of optimising its parameters. In fact, the typical solvent extraction can be time-consuming, with the extraction requiring up to 48 h and the need for constant monitoring of stability parameters. Additionally, an efficient SE is still wasteful and utilizes environmentally harmful solvents, as listed in the previous paragraph [128,129]. Recently, Churilov et al. [130] devised a new method of extraction for empty cage fullerenes, called the mechanical extraction method, where a rotating shaft is used to rotate and grind the fullerenes, thus breaking the van der Waals bonds between the fullerenes and insoluble carbon particles from the soot. This also helps in separating the soluble fullerenes from the insoluble fullerenes in the solvent, thus efficiently separating the two in the extraction process. The combination of the mechanical influence with the filtration led to a massive reduction in the time taken for extraction (~15 min) compared to the traditional Soxhlet extraction method, whilst maintaining a similar resultant weight percentage of fullerene-to-soot-extraction ratio. The same group went on to demonstrate the efficiency of the method with higher cage empty fullerenes and EMFs [131]. They showed that the mechanical extraction method could extract 0.4 wt% higher weight fullerenes, which consisted of about 25% EMFs. Although the authors showed this methodology to work well with both EMFs and empty cages, we believe that it is worth using the mechanical extraction method and Soxhlet extraction to efficiently extract empty cage fullerenes and subsequently EMFs, respectively. Moreover, improvements can be further made in the mechanical extraction process by introducing heating in the system. This could lead to increased breaking of the fullerene-carbon particle bond and their increased solubility in the solvent.

## 5. Conclusions

In conclusion, we summarized some of the important parameters needed to optimize the synthesis of endohedral metallofullerenes. It is important to increase the yield of these nanomaterials in order to carry out in-depth investigation of their synthesis mechanism as well as to ensure a sustainable use for future electronic and opto-electronic devices. Challenges still exist in scaling-up the synthesis of individual fullerene molecules and optimizing separation of individual fullerenes from the obtained soot. While the conventional arc-discharge method, based on the vaporization of graphite composites, seems to have some limitations, the doping of the rods, despite being a more tedious and uncontrollable process, can be further expanded and provide new insights and pathways. New doping agents and optimized reaction conditions can be developed for further studying the role of

different parameters during synthesis. Our ongoing research has a final aim of mapping the temperature and pressure conditions during the arc-discharge process by utilizing tailor-made arc-discharge reactors and carefully controlling and recording the reaction parameters and conditions. Incorporation of thermocouples inside the reactor to map the temperature during the synthesis will help in understanding the reaction conditions that are best suited for optimum synthesis yield of the endohedral fullerenes of interest [132]. This mapping is important not only for the scaled-up production of EMFs but also for the synthesis of new clusters, presenting either new properties or combining optical, optoelectronic, electrochemical, and magnetic properties. Examples of such materials are EMFs that incorporate erbium with gadolinium or dysprosium in different ratios, with the former providing the optical properties and the latter providing the magnetic behaviour. New advances, such as the construction of atomic clocks with $N@C_{60}$ [133], can provide new directions for the applications of endohedral fullerenes. In particular, EMFs of both well-known (such as $Y@C_{82}$, $Sc@C_{82}$, and $La@C_{82}$) or newly developed materials with different electron and nuclear spin numbers (such as phosphorous-containing endohedral fullerenes) could potentially be used for atomic clock applications. The incorporation of machine learning (ML) and artificial intelligence (AI) approaches for the synthesis of EMFs could revolutionize the industrial-scale production of these molecules by defining optimal process parameters and complementing traditional techniques such as the Design of Experiments (DoE) approach [134], favoured by engineers for many decades. This is a direction in which we plan to endeavour in the near future.

Finally, a combination of optical and paramagnetic properties [135–137] can also lead to new carbon nanotube–EMF peapods or EMF–graphene hybrid nanomaterials [138] with novel properties. Even in organic photovoltaics, fullerenes have recently been incorporated in ternary systems [139]. The incorporation of ternary systems including EMFs and non-fullerene acceptors could provide further avenues for higher-efficiency solar cells. Thus, the science and technology of EMFs will be a viable research topic for years to come.

**Author Contributions:** Conceptualization, S.S. and K.P.; methodology, S.S.; software, S.S.; writing—original draft preparation, S.S., P.D. and K.P.; writing—review and editing, S.S., K.S., S.K., P.D. and K.P. All authors have read and agreed to the published version of the manuscript.

**Funding:** This research was funded by EPSRC grant number EP/K030108/1.

**Acknowledgments:** S.S. would like to thank the Schmidt Science Fellows for financial support. K.P. would like to acknowledge EPSRC for grant support (EP/K030108/1).

**Conflicts of Interest:** The authors declare no conflict of interest.

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
