# Peer review of "Process Parameter Optimisation for Endohedral Metallofullerene Synthesis via the Arc-Discharge Method"

_inorganics, doi:10.3390/inorganics12020038_

Round 1

Reviewer 1 Report

Comments and Suggestions for Authors

The review titled “Process Parameters Optimisation for Endohedral Metallofullerene Synthesis via the Arc-Discharge Method” by Sinha et al. provides a detailed examination of the synthesis methodologies for endohedral metallofullerenes (EMFs) through the arc-discharge method. The authors effectively review various methodologies, offering valuable insights for readers interested in EMF synthesis. The inclusion of figures and tables showing key synthesis parameters and their impact on EMF yield could improve the accessibility of EMF. Overall, the paper is well-structured, and the authors have made a notable contribution to the understanding of EMF synthesis. I will suggest the publication of this review on Inorganics after addressing some minor issues.

(1) To further strengthen the scientific rigor of the review, a more in-depth description of the challenges and potential strategies for overcoming the low yield of EMFs will be recommended.

(2) Encouraging the authors to elaborate on the practical applications of the future directions for EMFs, which would provide a more objective assessment of the review’s impact.

(3) Some format mistakes:

a. In lines 53-54 and Figure 1, the point groups, such as D3 and C2v, should be modified with D, C, and v in italics.

b. Line 74, correct naming for Sc4O3@C80 is Sc43-O)3@Ih(7)-C80.

c. Line 79, an additional “-” occurs after reference 19.

d. Line 80, the n for C2n should be written in italics.

e. In lines 106 and 489 and Table 1, there is a missing space between the number and unit.

f. Line 493, the correct naming for the spin is I = 2/7.

g. Line 527, there is a missing space between “and” and “Y@C74(CF3)3”.

I will strongly suggest rechecking the review, especially for the formats, such as the point groups, naming, units, and missing spaces.

Comments on the Quality of English Language

No

Author Response

Letter of response to the reviewer’s comments:

Reviewer 1:

The review titled “Process Parameters Optimisation for Endohedral Metallofullerene Synthesis via the Arc-Discharge Method” by Sinha et al. provides a detailed examination of the synthesis methodologies for endohedral metallofullerenes (EMFs) through the arc-discharge method. The authors effectively review various methodologies, offering valuable insights for readers interested in EMF synthesis. The inclusion of figures and tables showing key synthesis parameters and their impact on EMF yield could improve the accessibility of EMF. Overall, the paper is well-structured, and the authors have made a notable contribution to the understanding of EMF synthesis. I will suggest the publication of this review on Inorganics after addressing some minor issues.

Comment (1): To further strengthen the scientific rigor of the review, a more in-depth description of the challenges and potential strategies for overcoming the low yield of EMFs will be recommended.
Response: We have now added more text throughout the section, as well as in the conclusions section to further discuss the challenges and potential strategies to overcome the low yield of EMFs. In particular, we also recommend using the new advances made in deep learning to learn from the existing wealth of literature and understand the mechanism of fullerene synthesis during the arc-discharge process. This would also assist in developing concrete strategy for overcoming the low yield of EMFs.

--------------------------------------------------------------------------------------------------------------------

Comment (2): Encouraging the authors to elaborate on the practical applications of the future directions for EMFs, which would provide a more objective assessment of the review’s impact.
Response: We have now added more text in the Conclusions section and in the appropriate section in the paper on the practical future directions for EMFs.

--------------------------------------------------------------------------------------------------------------------

Comment (3) Some format mistakes:

  1. In lines 53-54 and Figure 1, the point groups, such as D3 and C2v, should be modified with D, C, and v in italics.
  2. Line 74, correct naming for Sc4O3@C80 is Sc4(μ3-O)3@Ih(7)-C80.
  3. c. Line 79, an additional “-” occurs after reference 19.
  4. Line 80, the n for C2n should be written in italics.
  5. In lines 106 and 489 and Table 1, there is a missing space between the number and unit.
  6. Line 493, the correct naming for the spin is I = 2/7.
  7. Line 527, there is a missing space between “and” and “Y@C74(CF3)3”.

I will strongly suggest rechecking the review, especially for the formats, such as the point groups, naming, units, and missing spaces.

Response: We have now corrected these mistakes. We have also checked the document for all other formatting errors, naming, units and missing spaces. Thank you for bringing it to our attention.

------------------------------------------------------------------------------------------------------------------

Reviewer 2 Report

Comments and Suggestions for Authors

In this short review article, the authors systematically introduce and summarize the main reaction parameters affecting the experimental yield of EMFs for the popular arc discharge technology. The review is timely and important to guide the future study of EMFs because the main bottleneck in this field is still their rather low yield although after more than three decades of first discovery. The manuscript is thus strongly recommended to be accepted after further addressing a few issues.  

1) Could the authors give a table comparing the pro and cons of different methods for synthesizing EMFs? This is important for readers to understand the motivation of this work.

2) Besides giving the cage symmetry, it would be much better to name different fullerene isomers in the texts and figures (e.g., Figure 1) using the well-accepted spiral codes proposed by Fowler and Manolopoulos. Here, IPR and non-IPR isomers can be numbered separately as done in many fullerenes and EMFs literature.

3) Can they give some discussions or insights why are most transition metals in the periodic table scarcely obtained as EMFs? What is the bottleneck? And how to make a step towards this direction?

Author Response

Letter of response to the reviewer’s comments:

Reviewer 2:

In this short review article, the authors systematically introduce and summarize the main reaction parameters affecting the experimental yield of EMFs for the popular arc discharge technology. The review is timely and important to guide the future study of EMFs because the main bottleneck in this field is still their rather low yield although after more than three decades of first discovery. The manuscript is thus strongly recommended to be accepted after further addressing a few issues. 

Comment 1) Could the authors give a table comparing the pro and cons of different methods for synthesizing EMFs? This is important for readers to understand the motivation of this work.

Response: We have now added a table comparing the pros and cons of different methods that are usually employed for synthesizing EMFs.

-------------------------------------------------------------------------------------------------------------------

Comment 2) Besides giving the cage symmetry, it would be much better to name different fullerene isomers in the texts and figures (e.g., Figure 1) using the well-accepted spiral codes proposed by Fowler and Manolopoulos. Here, IPR and non-IPR isomers can be numbered separately as done in many fullerenes and EMFs literature.

Response: We have now also updated the naming of the fullerene isomers by using the well-accepted spiral codes proposed by Fowler and Manolopoulos, such as replacing Sc4O3@C80 with Sc4(μ3-O)3@Ih(7)-C80.

--------------------------------------------------------------------------------------------------------------------

Comment 3) Can they give some discussions or insights why are most transition metals in the periodic table scarcely obtained as EMFs? What is the bottleneck? And how to make a step towards this direction?

Response: Transition metals are scarcely obtained as EMFs due to the interplay between reactivity, thermodynamic stability and synthetic challenges. owing to their high reactivity. Transition metals tend to form compounds with other elements readily, making it challenging to isolate them within the closed, stable environment of a fullerene cage. However, in the past decade, many transition metals have also been incorporated inside the fullerenes. We have now updated the section.

----------------------------------------------------------------------------------------------------------------

Reviewer 3 Report

Comments and Suggestions for Authors

The review article by Sinha and Porfyrakis et al. summarized the progress of parameter optimization for endohedral metallofullerene synthesis via the arc-discharge method. Endohedral metallofullerene synthesis is an old topic however with several key questions (such as the reaction mechanism mentioned in the article) unanswered until now. I believe it is important to make such a summary to guide the development of endohedral metallofullerene synthesis. The authors may consider the following concerns before publishing.

The authors summarized the development from the following aspects of reaction parameters and pathways: anode composition, ionization potential of metals, effect of catalyst, effect of helium pressure, solvent extraction. In-situ doping, and methane and ammonia addition: effect of reactive gases were discussed as a separate part with the title of “Chemically Activated Arc Discharge” I feel that the separated part should be better combined with the part of “Reaction Parameters and Pathways”, and the “Solvent Extraction” should be moved out of the part of “Reaction Parameters and Pathways” because it is not the reaction parameters and pathways anymore.

The choice of six different types of isomers of C80 in Figure 1 seems problematic: firstly, how to differentiate the two C2v isomers should be described clearly; secondly, why 6 isomers, not all 7 IPR isomers of C80?

As I know, P inside a fullerene cage still needs solid evidence (ref. 18), mentioning this in this review seems unnecessary.

The background of Figure 2 may be changed to improve the visibility.

Line 102: the authors claimed, "For instance, carbon cages of EMFs have different isomeric structures and show higher solubility in polar solvents.” I think that the claim may be misleading.

Line 161-164: “Furthermore, it was also observed that these metal clusters had motions and trajectories inside and the cage, which also changed positions depending on the temperature.” I think that citing the theoretical and spectroscopic results (refs. 54 and 55) on the metal motions inside the cage is not convincing enough, more convincing, and accurate single crystal X-ray diffraction results have been reported recently, such as, Nat. Commun. 2019, 10, 571; J. Am. Chem. Soc. 2021, 143, 612-616.

Author Response

Letter of response to the reviewer’s comments:

Reviewer 3:

The review article by Sinha and Porfyrakis et al. summarized the progress of parameter optimization for endohedral metallofullerene synthesis via the arc-discharge method. Endohedral metallofullerene synthesis is an old topic however with several key questions (such as the reaction mechanism mentioned in the article) unanswered until now. I believe it is important to make such a summary to guide the development of endohedral metallofullerene synthesis. The authors may consider the following concerns before publishing.

The authors summarized the development from the following aspects of reaction parameters and pathways: anode composition, ionization potential of metals, effect of catalyst, effect of helium pressure, solvent extraction. In-situ doping, and methane and ammonia addition: effect of reactive gases were discussed as a separate part with the title of “Chemically Activated Arc Discharge” I feel that the separated part should be better combined with the part of “Reaction Parameters and Pathways”, and the “Solvent Extraction” should be moved out of the part of “Reaction Parameters and Pathways” because it is not the reaction parameters and pathways anymore.

Response: We have now moved the sections such that the chemically activated arc discharge is under the parameters section and solvent extraction is a separate section. Thank you for the insight.

---------------------------------------------------------------------------------------------------------------------

The choice of six different types of isomers of C80 in Figure 1 seems problematic: firstly, how to differentiate the two C2v isomers should be described clearly; secondly, why 6 isomers, not all 7 IPR isomers of C80?

Response: The two isomers are actually C2v and C2v’. We realized that the apostate on the other 2v’ was not easily visible, so we have now updated the figure to easily identify it as C2v’. We only showed the 6 isomers because they are experimentally known to be the most stable. Unlike C60, where Ih isomer is the most stable, in this case, C80 experiments a Jahn-Teller distortion and as a result arise species with lower energy and less symmetry. We have now updated the text to mention that these are “the six most stable isomers” and also included the citation.

---------------------------------------------------------------------------------------------------------------------

As I know, P inside a fullerene cage still needs solid evidence (ref. 18), mentioning this in this review seems unnecessary.

Response: We have now removed it from the review. Thank you for bringing it to our attention.

---------------------------------------------------------------------------------------------------------------------

The background of Figure 2 may be changed to improve the visibility.

Response: Thank you for the comment, we have now changed the background of Figure 2 to improve the visibility.

---------------------------------------------------------------------------------------------------------------------

Line 102: the authors claimed, "For instance, carbon cages of EMFs have different isomeric structures and show higher solubility in polar solvents.” I think that the claim may be misleading.

Response: We wrote that sentence with particular metallofullerenes in mind. However, we do recognize that this claim may be misleading. We have now modified the statement to reflect different solubility to different solvents, instead of claiming higher or lower solubility in any particular ones.

-------------------------------------------------------------------------------------------------------------------

Line 161-164: “Furthermore, it was also observed that these metal clusters had motions and trajectories inside and the cage, which also changed positions depending on the temperature.” I think that citing the theoretical and spectroscopic results (refs. 54 and 55) on the metal motions inside the cage is not convincing enough, more convincing, and accurate single crystal X-ray diffraction results have been reported recently, such as, Nat. Commun. 2019, 10, 571; J. Am. Chem. Soc. 2021, 143, 612-616.

Response: Thank you for bringing this to our attention. We have now updated the text and the reference for this section including these two citations.

---------------------------------------------------------------------------------------------------------------------